# Fillet Fish Fortified with Algal Extracts of *Codium tomentosum* and *Actinotrichia fragilis*, as a Potential Antibacterial and Antioxidant Food Supplement

**DOI:** 10.3390/md20120785

**Published:** 2022-12-17

**Authors:** Mohamed S. M. Abd El Hafez, Sarah H. Rashedy, Neveen M. Abdelmotilib, Hala E. Abou El-Hassayeb, João Cotas, Leonel Pereira

**Affiliations:** 1National Institute of Oceanography and Fisheries, NIOF, Cairo 11516, Egypt; 2Center of Excellence for Drug Preclinical Studies (CE-DPS), Pharmaceutical and Fermentation Industries Development Center (PFIDC), City of Scientific Research and Technological Applications (SRTA-City), New Borg El-Arab City 21934, Egypt; 3Department of Food Technology, Arid Lands Cultivation Research Institute (ALCRI), City of Scientific Research and Technological Applications (SRTA-CITY), New Borg El-Arab City 21934, Egypt; 4MARE—Marine and Environmental Sciences Centre, Department of Life Sciences, University of Coimbra, Calçada Martim de Freitas, 3000-456 Coimbra, Portugal

**Keywords:** marine natural products, seaweed, HPLC, DPPH, cytotoxicity

## Abstract

With respect to the potential natural resources in the marine environment, marine macroalgae or seaweeds are recognized to have health impacts. Two marine algae that are found in the Red Sea, *Codium tomentosum* (Green algae) and *Actinotrichia fragilis* (Red algae), were collected. Antibacterial and antioxidant activities of aqueous extracts of these algae were evaluated in vitro. Polyphenols from the extracts were determined using HPLC. Fillet fish was fortified with these algal extracts in an attempt to improve its nutritional value, and sensory evaluation was performed. The antibacterial effect of *C. tomentosum* extract was found to be superior to that of *A. fragilis* extract. Total phenolic contents of *C. tomentosum* and *A. fragilis* aqueous extract were 32.28 ± 1.63 mg/g and 19.96 ± 1.28 mg/g, respectively, while total flavonoid contents were 4.54 ± 1.48 mg/g and 3.86 ± 1.02 mg/g, respectively. Extract of *C. tomentosum* demonstrates the highest antioxidant activity, with an IC_50_ value of 75.32 ± 0.07 μg/mL. The IC_50_ of L-ascorbic acid as a positive control was 22.71 ± 0.03 μg/mL. The IC_50_ values for inhibiting proliferation on normal PBMC cells were 33.7 ± 1.02 µg/mL and 51.0 ± 1.14 µg/mL for *C. tomentosum* and *A. fragilis*, respectively. The results indicated that both algal aqueous extracts were safe, with low toxicity to normal cells. Interestingly, fillet fish fortified with *C. tomentosum* extract demonstrated the greatest overall acceptance score. These findings highlight the potential of these seaweed species for cultivation as a sustainable and safe source of therapeutic compounds for treating human and fish diseases, as well as effective food supplements and preservatives instead of chemical ones after performing in vivo assays.

## 1. Introduction

Marine natural products are extremely useful substances for use in food additives and drug discovery as a result of their broad variety of bioactivities [1,2,3]. Marine organisms use powerful secondary metabolites to prevent the growth of invading neighbors and attract food. Such survival conditions stimulate the production of an extremely abundant variety of biologically active substances [4,5,6,7]. Secondary metabolites isolated from different alga are playing an important role as lead components, natural medicine or nutraceuticals in drug discovery research and pharmaceutical industries and natural food preservatives [8,9]. Recently, due to the resistance of different pathogenic bacteria and pests to antibiotics and insecticidal agents, finding new active components against these health and environmental problems is one of the major areas of medical and agricultural research. The peculiarity of the marine environment and the high biological activity of marine natural products make algal metabolites a fascinating source for finding new antimicrobial and insecticidal compound [10,11,12].

Seaweeds are large and diverse groups of plants that are rich in active metabolites and a source of novel ingredients for functional foods. Nutritional studies on seaweeds indicate that brown, green and red seaweeds possess good nutritional quality and could be used as an alternative source of dietary fiber, protein and minerals. Seaweeds are also considered as a source of bioactive compounds as they are able to produce a great variety of secondary metabolites, characterized by a wide range of biological activities such as antimicrobial, anti-inflammatory, antiviral as well as antitumor activities [13,14,15]. Moreover, many studies show that some algae extracts display substantial antioxidant activities. Antioxidant substances in seaweeds contribute to the endogenous defense mechanism against external stressful conditions. Antioxidant properties of some red, brown and green algae extracts have shown that they vary in proportion to the content of antioxidative compounds [16,17].

Algae are a unique raw material for the production of a number of substances with a wide range of useful properties. Their composition is characterized by the specific content of minerals, pigments, lipids, polyphenols, proteins, amino acids, cellulose, polysaccharides, etc. One of the most significant groups of compounds that determines the biomedical importance of marine algae is polyphenols. The largest proportion of phenolic compounds in green and red algae is represented by bromophenols, phenolic acids and flavonoids [18]. Seaweed phenolic compounds are metabolites that are scientifically defined as molecules with hydroxylated aromatic rings [19,20,21]. These phytochemicals have a diverse chemical structure, ranging from simple moieties to large molecular polymers. The shikimate or acetate pathway is the main metabolic pathway which generates these phytochemicals [22,23,24]. Bromophenols, flavonoids, phenolic acids, phenolic terpenoids and mycosporine-like amino acids account for the majority of phenolic chemicals found in green and red seaweeds [25,26,27,28]. These compounds are classified as secondary metabolites because they are protective agents that are produced in response to various stimuli and serve as seaweed defense mechanisms against herbivory and UV exposure [29].

The majority of phenolic compounds have anti-diabetic, anti-inflammatory, antimicrobial, antiviral, anti-allergic, antioxidant, antiphotoaging, antipruritic, hepatoprotective, hypotension, neuroprotective and anticancer properties [19,28,30,31,32,33,34,35,36,37,38,39,40,41]. Given their many bioactivities, seaweeds are ideal candidates for the creation of goods or components utilized in commercial applications such as medicines, cosmetics, functional foods and even bioactive food packaging films to preserve food quality [38,40,42,43,44]. Because of their structural similarities and proclivity to react with other chemicals, phenolic compounds are extremely difficult to extract quantitatively on an industrial scale [19]. Thus, seaweed extracts can be a solution to reduce costs and attain economic feasibility at a large scale.

Therefore, the aim of the present study was to determine the polyphenol content and antioxidant and antibacterial activities of *Codium tomentosum* (Chlorophyta) and *Actinotrichia fragilis* (Rhodophyta) from the Red Sea (Egypt). Furthermore, we fortified fillet fish with these algal extracts in an attempt to improve nutritional values and food safety while maintaining a pleasant taste.

## 2. Results and Discussion

In the present study, the aqueous extracts of *C. tomentosum* and *A. fragilis* were investigated for their antioxidant and antimicrobial activity, and their polyphenol contents were also investigated to provide clarification of the chemical constituents. Macroalgae are ecologically and biologically important natural sources. They are an important basis for therapeutically useful substances. Due to their biological and chemical variations, the marine environment may be a source of novel types of antimicrobial agents and biologically active compounds.

### 2.1. Antibacterial Activity and MIC Determination

Antibacterial activities expressed as inhibition zone diameters of algal aqueous extracts against the tested bacterial strains are shown in Table 1. The antibacterial effects of *C. tomentosum* aqueous extract were found to be superior to those of *A. fragilis* extract in the experiment.

The minimum inhibitory concentration (MIC) is an essential factor that assesses microorganism resistance and sensitivity to specific substances. The MIC of *C. tomentosum* and *A. fragilis* against the bacterial strains is shown in Table 2.

Seaweeds are potential renewable resources of bioactive compounds with diverse beneficial effects. Several studies reported that bioactive secondary metabolites isolated from various seaweed exhibit potential to be used as antimicrobial molecules. Nevertheless, species belonging to the genus *Codium* have been the least investigated among all members of Chlorophyta for their biological activities and their possible use in food and biomedical applications. Aqueous extracts of *C. tomentosum* and *A. fragilis* were subjected to antibacterial assay against a wide array of bacterial pathogens. The selected bacteria are among the most common causes of foodborne and infectious diseases. They showed an extended spectrum of inhibitory activity against all the bacterial pathogens [45].

In addition, there are reported studies that describe the antibacterial capability (derived from secondary and primary metabolites) of seaweeds against medically important pathogenic bacteria. As previously shown in many studies, marine macroalgae have antimicrobial components that inhibit the growth of some bacteria. It has also been reported that the efficacy of macroalgae extracts against microorganisms is mostly influenced by factors such as location and seasonality. Another study of macroalgae showed a high percentage of species with antimicrobial activity, 73% in the case of Chlorophyta (green algae), 69% in Rhodophyta (red algae) and 53% in Phaeophyceae (brown algae) [46]. Rajauria et al. used aqueous phenolic extract to demonstrate that *Himanthalia elongata* have antimicrobial activity against *Listeria monocytogenes* ATCC 19115, *Salmonella abony* NCTC 6017, *Enterococcus faecalis* ATCC 7080 and *Pseudomonas aeruginosa* ATCC 27853 [47].

The findings of the present investigation are analogous to those observed by Elkhateeb et al. [48], who reported that the crude extract of *C. tomentosum* showed strong antimicrobial activity against two Gram-positive bacteria, *Staphylococcus aureus* ATCC 6538 and *Bacillus subtilis* ATCC 6633, as well as two Gram-negative bacteria, *Escherichia coli* ATCC 19404 and *Vibrio alginolyticus* MK 170250.

However, in other studies such as that of Koz et al. [46], antibacterial activity of *Codium fragile* extract against *E. aerogenes*, *E. coli* and *B. subtilis* was observed. Antibacterial activity of *C. intricatum* extract in opposition to *S. aureus* and MRSA is similar to that reported from methanol extracts of *Codium* species (*C. tomentosum*, *C. tomentosum*, *C. dichotomum* and *C. fragile*), where high inhibitory activity was noted [45]. Pasdaran et al. [10] reported the antibacterial activity of *A. fragilis* volatile oil against *Pseudomonas aeruginosa*, *E. coli* and *Staphylococcus aureus*. Salem et al. [49] reported the antibacterial activity of *A. fragilis* extracts against *E. coli*, *S. aureus*, *E. feacalis*, *Salmonella* sp., *B. cereus* and *P. aeruginosa*. Alghazeer et al. [50] reported that extracts from *C. tomentosum* possess in vitro antibacterial activity against eight bacterial strains, namely *S. aureus*, *B. subtilis*, *Bacillus* spp., *S. epidermidis*, *S. typhi*, *E. coli*, *P. aeruginosa* and *klebsiella* spp., similar to the findings of this study.

The sensitivity of a specific kind of bacteria to the activity of bioactive substances found in the algal extracts is attributed to the difference in structure and composition of the cell walls. Gram-positive bacteria are marked by dense peptidoglycan in the outer layer of the cell wall, while Gram-negative bacteria have a composite, multilayered cell wall structure that makes the entry of bioactive compounds more difficult [51].

### 2.2. Antioxidant Activity

Phenolic compounds are key antioxidant and antibacterial agents with numerous benefits for disease prevention and human health. Flavonoids are natural substances with a polyphenolic structure; as a result, they have antibacterial and antioxidant activity and can help prevent illnesses including Alzheimer’s disease, cancer and atherosclerosis [52]. Total phenolic and flavonoid contents of *Codium tomentosum* and *Actinotrichia fragilis* are shown in Table 3.

Total phenolic contents of *C. tomentosum* and *A. fragilis* aqueous extract were 32.28 ± 1.63 mg/g and 19.96 ± 1.28 mg/g of extract, respectively, while TFCs were 4.54 ± 1.48 mg/g and 3.86 ± 1.02 mg/g of extract, respectively. TFC’s significance comes mostly from its redox characteristics that might account for its antibacterial and antioxidant activities against a variety of microorganisms and free radicals. Furthermore, the action of TFC as an antioxidant agent is closely related to the hydroxyl groups in its structures, which are responsible for scavenging lipid peroxy-radicals, singlet oxygen, superoxide anion and free radical stabilization [53,54,55].

The DPPH is a significant metric for determining an extract’s antioxidant activity. The IC_50_ is the extract concentration needed to scavenge 50% of the DPPH radicals. The IC_50_ values of *C. tomentosum* and *A. fragilis* aqueous extracts are presented in Table 4. The better the antioxidant properties, the smaller the IC_50_ value. According to the results, *C. tomentosum* aqueous extract demonstrates the highest antioxidant activity, with an IC_50_ value of 75.32 ± 0.07 μg/mL. The IC_50_ of L-ascorbic acid as a positive control was 22.71 ± 0.03 μg/mL.

Basically, antioxidants delay oxidation and reduce oxidative damage, which is a significant causative factor in the development of many chronic diseases. Oxidative stress is an important factor in the pathogenesis of various diseases such as atherosclerosis, cancer and aging. Naturally generated reactive oxygen species can attack cell components and then exert several types of biological damage and oxidative stress [56,57,58]. Antioxidants protect against these reactions which occur in vital systems and increase shelf-life when added to lipids and lipid-containing foods [46,52]. Although synthetic antioxidants such as butylatedhydroxyanisole, butylatedhydroxytoluene and propyl gallate have been used for many years, they have started to be restricted in recent years because of their carcinogenicity. Thus, there is a gradual increase in the investigations to identify new natural antioxidants [46,52].

Seaweeds are considered to be important sources of antioxidants. The results from antioxidant activity screening in the extracts suggested that algae extracts have radical scavenging inhibitor activity. The findings of the present investigation are similar to those observed by Alghazeer et al. [16], who reported that *C. tomentosum* extracts possess antioxidant and antiproliferative activities which might be helpful in preventing or slowing the progress of various oxidative stress-related disorders. The findings of the present investigation are similar to those observed by Baskaran et al. [59], who reported that methanol crude extracts of *A. fragilis* contain different potential antioxidant compounds able to scavenge different types of free radicals. *Acanthophora specifera* flavonoid separation reveals a combination of chlorogenic acid (69.64%), caffeic acid (12.86%), vitexin-rahmnose (12.35%), quercetin (1.41%) and catechol (0.59%) [60]. The antioxidant activity of the flavonoid-enriched extract has been proven to be very high [61]

Phenolic compounds have exceptional antioxidant properties due to their capacity to function as chelating agents with reactive oxygen species, avoiding oxidative stress and cell damage [29,62]. As a result, scavenging of oxidants is critical for disease prevention, and phenolic compounds found in seaweeds are particularly helpful as a natural supply of antioxidant agents [52]. The study of Agregán et al. [63] demonstrated that aqueous polyphenolic extracts of *Ascophyllum nodosum*, *Bifurcaria bifurcata* and *Fucus vesiculosus* have antioxidant activity, which stabilize the canola oil oxidation level.

Nonetheless, synthetic ingredient limits in the food business may represent a tipping point for the use of seaweed compounds as safe replacements [64], since they also exhibit anti-microbial activity against key food spoilage and food pathogenic microorganisms [65]. Seaweed phenolic antioxidant extracts have been used to improve oxidative stability and to preserve or boost the intrinsic quality and nutritional content of foods [66,67]. The antioxidant potential is useful in the food industry not only for nutraceutical compounds on functional food products, where they are indisputably valuable for health improvement (as food supplements), but also to extend the shelf-life period when used in processed food (functional foods) [68,69]. Furthermore, the antibacterial activity of seaweed phenolics suggests that they can be valuable in the food industry [70].

### 2.3. Cytotoxicity Assay

The influence of various concentrations of *Codium tomentosum* and *Actinotrichia fragilis* aqueous extracts on PBMC viability are given in Table 5 and Table 6. The IC_50_ value for inhibiting proliferation on normal PBMC cells was 33.7 ± 1.02 µg/mL and 51.0 ± 1.14 µg/mL for *C. tomentosum* and *Actinotrichia fragilis*, respectively. The results indicated that both algal aqueous extracts were safe with low toxicity to normal cells and could be used as novel and effective food preservatives instead of chemical ones after performing in vivo assays, acting also as nutraceuticals [52].

The current study’s findings were consistent with those published by Alghazeer et al. [16], who reported that seaweeds rich in bioactive compounds may be used in anticancer drug research programs. However, further investigations are essential to reveal the molecular mechanisms of the anticancer activities of these algae.

### 2.4. HPLC Analysis

To identify bioactive compounds in *C. tomentosum* and *A. fragilis* aqueous extracts, high-performance liquid chromatography (HPLC) was used. The HPLC chromatogram of standards and phenolic compounds of *C. tomentosum* and *A. fragilis* aqueous extracts are shown in Table 7 and Figure 1.

The HPLC analysis results confirmed the presence of several polyphenolic compounds when compared to the HPLC standard chromatogram. These findings agreed with those of Tanna et al. [36], who revealed phenolic, flavonoid and amino acid compositions in some species of seaweeds that are promising as functional food ingredients or dietary supplements for daily intake.

HPLC analysis results of *C. tomentosum* and *A. fragilis* extracts confirmed the in vitro antioxidant and antimicrobial studies, which demonstrated their biochemical components. These active components are also considered the main reason for free radical scavenging activities of our target seaweeds, demonstrating the nutraceutical potential of these extracts and also highlighting them as a raw source of interesting natural compounds with pharmacological potential [52].

### 2.5. Sensory Evaluation of Fillet Fish Fortified with Algal Extracts

Regarding the bioactivity and biochemical potential of these two extracts, it is important to understand if they can be used as food additives and applied in the food industry [52,71]. Table 8 shows the sensory evaluation scores of fillet fish fortified with algal extracts. According to the findings, fillet fish fortified with algal extracts has been accepted as having high nutritional value for human diet. Interestingly, fillet fish fortified with *C. tomentosum* extract demonstrated the greatest overall acceptance score, as well as the highest antibacterial and antioxidant activity.

Water extracts were used because they are safer solvents than other organic solvents. Water extracts contain a variety of constituents, including polysaccharides and polyphenols, but we focused on polyphenols in this study because they are powerful antioxidants and antibacterial agents, as reported previously by several studies. This important observation will be made in subsequent work.

In this preliminary work, we treated the fillet fish with aqueous extracts mixed with spices immediately before frying in oil. Only sensory evaluation was performed on the fillet fish fortified with algal extracts in an attempt to improve its nutritional value in comparison to our standard fillet fish diets.

Our future studies will focus on fortifying fillet fish or yogurt with natural seaweed extracts as natural preservatives instead of chemical ones. pH values, titratable acidities (TAs), total soluble solids, water holding capacity (WHC), thiobarbituric acid reactive substances (TBARS), lipid oxidation, protein oxidation (modified DNPH carbonyl assay) and microbiological properties will be evaluated over 30 days of storage.

The constraints on synthetic components in the food business may be a tipping moment for the use of seaweed compounds as safe replacements [64], as they also exhibit antimicrobial properties against major food spoilage and pathogenic microbes [65]. Seaweed phenolic antioxidant extracts have been used to improve oxidative stability and to preserve or boost the intrinsic quality and nutritional content of foods [66,67].

The antioxidant potential is beneficial in the food business not only as nutraceutical compounds in functional food items, where they are undeniably valuable for health improvement (as food supplements), but also to extend the shelf-life period when included in processed foods (functional foods) [68,69]. Furthermore, the antibacterial properties of seaweed phenolics suggest that they may be valuable in the food business [70].

Furthermore, bromophenols enriched extract from *Ulva lactuca* and *Pterocladiella capillacea* were investigated as “marine flavour” agents in farmed fish and other aquatic organisms, because farming final products can differ in flavor from wild catches, and this can be incorporated as a feed ingredient or a seaweed bromophenol-enriched sauce [72].

## 3. Materials and Methods

### 3.1. Algae Collection

Two marine algae species, *Codium tomentosum* Stackhouse, a green marine macroalgae (phylum Chlorophyta, class Ulvophyceae), and *Actinotrichia fragilis* Forsskal, Borgesen, a red marine macroalgae (phylum Rhodophyta, class Florideophyceae), were harvested in summer 2021 from the tidal zone of the Red Sea shore at Hurghada, Egypt, between latitude 27°28.15′ N and longitude 33°77.13′ E (Figure 2). The seaweed species were identified under a stereo microscope according to their morphological characteristics with taxonomic references following the descriptions of Aleem [73]. The two macroalgal species were collected randomly from the site in the summer season. Because of the large amount of collected algae used in the extraction, no reproductive stage was recorded.

### 3.2. Preparation of Algal Extracts

About 0.5 kg of each alga was washed with water, shade-dried and powdered in a mixer grinder. Hot water (80 °C) was used to extract the compounds from the samples (1:2, *w*/*v*) for one hour before separating the filtrate and residue. The extraction was repeated three times under similar conditions. Last, freeze-drying the filtrate yielded powdered algae aqueous extracts. The extracts were then kept at −20 °C for subsequent study.

### 3.3. Antibacterial Activity and MIC Determination

Agar well diffusion tests described by Abd El Hafez et al. [5] and Kadaikunnan et al. [74] were applied to detect the action of the extracts against the reference pathogenic strains—Gram-negative bacteria *Klebsiella pneumonia* (ATCC700603) and *E. coli* (ATCC25922), and Gram-positive bacteria *Staphylococcus aureus* (ATCC6538) and *Streptococcus pyogenes* (EMCC1772)—obtained from City of Scientific Research and Technological Applications, by using 100 μL of the inoculums (1 × 10^8^ CFU/mL). Using descending concentrations, the minimum inhibitory concentration (MIC) of algal extracts toward the reference strains was established. In sterile saline, growing culture suspensions of the reference pathogenic strains were prepared and adapted to a concentration of 106 cells/mL. The reference pathogenic strain suspension was inoculated onto a plate of nutrient agar (Oxoid, U.K.) and left to dry for 15 min at room temperature. The algal extracts were serially diluted, and 100 μL of each dilution was independently applied to each well. The plates were incubated for 24 h at 37 °C before the MIC values were recorded. For each culture algal extract, the test was performed in triplicate.

### 3.4. Determination of Total Phenolic Contents

The total phenolic content was determined using the Folin–Ciocalteu reagent following [75] using gallic acid as a reference. Each sample was evaluated in three replicates. A 0.1 mL aliquot of Folin–Ciocalteu reagent was added to 0.1 mL of reconstituted extract. Then, 2.0 mL saturated sodium carbonate (7%) was added to the mixture after 15 min. For 30 min, the mixture was left at room temperature. TPC was determined using a spectrophotometer at 760 nm with gallic acid as a reference. The linear regression equation generated from the standard gallic acid calibration curve was used to calculate TPC as milligrams of gallic acid equivalent per gram sample. Each sample was evaluated in triplicate.

### 3.5. Determination of Total Flavonoid Contents

The total flavonoid contents were determined using a colorimetric method described by Sakanaka et al. [76]. First, 1.0 mL of the sample and 4.0 mL of water were mixed in a flask. Then, 0.75 mL of 5% sodium nitrite and 0.150 mL of 10% aluminum chloride were added to the mixture. After 5 min at room temperature, 0.5 mL of 1 M sodium hydroxide was added. A UV/VIS spectrophotometer was used to measure the absorbance at 510 nm. The results were expressed as milligram catechol equivalent per gram sample from the standard catechol calibration curve.

### 3.6. Antioxidant Potentials and DPPH Radical Scavenging Activity

The extracts’ free radical scavenging activity was determined using the DPPH method described by Catarino et al. [77]. Ascorbic acid was used as a standard with different ranges (10–100 μg/mL) to generate the standard curve. The absorbance was measured at 517 nm using a spectrophotometer. The activity of DPPH radical scavenging was determined as mg ascorbic acid equivalent (AAE)/g dry sample. The percentage of DPPH radical scavenging activity was calculated using the following Equation (1):(1)$$\mathrm{DPPH}\;\mathrm{radical}\;\mathrm{scavenging}\;\mathrm{activity}\;\left(\%\;\mathrm{inhibition}\right)=\frac{{\mathrm{Abs}}_{\mathrm{control}\;}-{\mathrm{Abs}}_{\mathrm{sample}}\;}{{\mathrm{Abs}}_{\mathrm{control}\;}}\times 100\;$$

### 3.7. HPLC Analysis

The phenolic profile of algal aqueous extracts was screened using an HPLC (Agilent 1260 series). An Eclipse C18 column with (4.6 mm × 250 mm i.d., 5 μm) was used and its temperature was kept constant at 40 °C. The mobile phase was water (A) and 0.05% trifluoroacetic acid in acetonitrile (B) and the volume of injection was 5 μL. The flow rate was 0.9 mL/min and the multi-wavelength detector was monitored at 280 nm.

### 3.8. Cytotoxicity Assay

In a 96-well tissue culture plate, the IC_50_ value of algal extracts on peripheral blood mononuclear cells (PBMCs) was defined using the neutral red cytotoxicity assay [78,79]. Extract concentrations of 250, 125, 62.5, 31.2, 15.6 and 7.8 µg/mL were obtained using double-fold serial dilutions. The cells in the wells were individually treated with various concentrations of extract, 150 µL in every well except for the blank. The following Formula (2) was utilized to determine the percentage of cytotoxicity (inhibition):(2)$$\%\;\mathrm{Inhibition}=\frac{O.D\;\mathrm{Control}-O.D\;\mathrm{Treatment}}{O.D\;\mathrm{Control}}$$

### 3.9. Sensory Evaluation of Fillet Fish Fortified with Algal Extracts

A total of 1 kg of fillet fish was treated by adding 1% of powdered algae aqueous extracts (*C. tomentosum* and *A. fragilis*) and then fried in oil. The samples were divided into four treatments: Treatment 1 (T1) as control (without extracts), Treatment 2 (T2) with 1% of *C. tomentosum* extract, Treatment 3 (T3) with 1% of A. fragilis extract and Treatment 4 (T4) as a mix with 0.5% of *C. tomentosum* and 0.5% of *A. fragilis* extracts.

The sensory evaluation panel included 15 members from the Food Technology Department and other departments, Arid Lands Cultivation Research Institute (ACRI), City of Scientific Research and Technological Applications (SRTA-City). The panel members were asked to observe some items (appearance, consistency, tenderness, flavor and overall eating quality) and give scores from 1 to 7 (excellent 7; very good 6; good 5; medium 4; fair 3; poor 2; very poor 1).

### 3.10. Statistical Analysis

All data collected were analyzed using one-way ANOVA, Fisher’s grouping test Minitab^®^ (Version 16) software, n = 3, and *p* < 0.05 revealed a significant difference. Significant means were compared by Duncan’s post hoc multiple comparison test.

## 4. Conclusions

It is well established that seaweeds contain various phytochemicals with diverse biological activities, are considered safe for human consumption and have significant health benefits. The current study provides approximate data on the phytochemical constituents of marine seaweeds *Codium tomentosum* and *Actinotrichia fragilis* aqueous extracts for use as food supplements fortified in fillet fish. Bioactive compounds found in seaweeds are awaiting a significant breakthrough to be used as natural antibacterial and antioxidant supplements in various food products. Moreover, these seaweed extracts have antibacterial activity against bacterial infections, lending credence to their traditional use and implying a future role for these seaweeds in combating microbial populations. The nutritional and physiological benefits of seaweeds suggest that they may be introduced to our regular diets as natural preservatives to promote health and immune system function, as well as enhance new pharmaceutical and food applications.

These results demonstrate that *C. tomentosum* aqueous extract can be further exploited. There is a need to advance our understanding of this interesting extract’s impact after fortified fillet consumption to understand if the bioactivity is maintained, improved or lowered.

## Figures and Tables

**Figure 1 marinedrugs-20-00785-f001:**
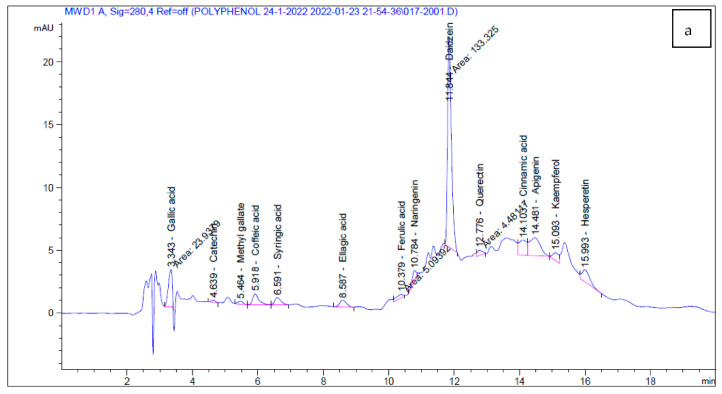
HPLC chromatograms of (**a**) *Codium tomentosum* extract, (**b**) *Actinotrichia fragilis* extract and (**c**) standards.

**Figure 2 marinedrugs-20-00785-f002:**
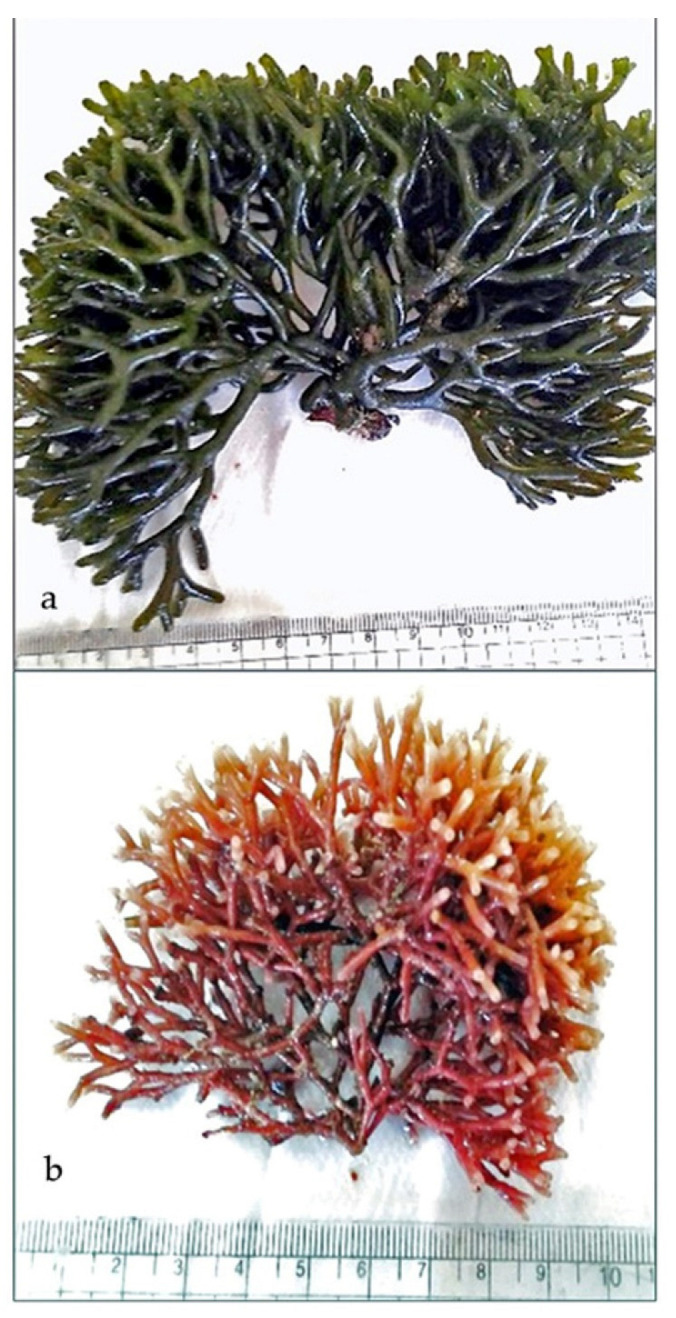
The collected marine algae: (**a**) *Codium tomentosum* (green algae) and (**b**) *Actinotrichia fragilis* (red algae).

**Table 1 marinedrugs-20-00785-t001:** Antibacterial potential of algal extracts.

Strains	Inhibition Zone Diameter (mm) ^1^	
	*Codium tomentosum* (Green Algae)	*Actinotrichia fragilis* (Red Algae)
Gram-positive bacteria		
*Staphylococcus aureus* (ATCC25923)	22 ± 0.04 ^a^	18 ± 0.05 ^b^
*Streptococcus pyogenes* (EMCC1772)	20 ± 0.01 ^a^	16 ± 0.01 ^b^
Gram-negative bacteria		
*Escherichia coli* (ATCC25922)	14 ± 0.08 ^a^	12 ± 0.05 ^b^
*Klebsiella pneumonia* (ATCC700603)	18 ± 0.02 ^a^	14 ± 0.07 ^b^

^1^ All results are expressed as the means ± standard deviation; n = 3. Diameter includes 5 mm well diameter. Different letters (^a^ and ^b^) indicate significant differences at *p* ≤ 0.05.

**Table 2 marinedrugs-20-00785-t002:** Minimum inhibitory concentration (MIC) of algal extracts against pathogenic bacteria indicated as a zone of inhibition for each concentration.

Strains	MIC (mg/mL) ^1^	
	*Codium tomentosum*	*Actinotrichia fragilis*
Gram-positive bacteria		
*Staphylococcus aureus* (ATCC25923)	0.4 ± 0.05 ^b^	0.8 ± 0.03 ^a^
*Streptococcus pyogenes* (EMCC1772)	0.4 ± 0.04 ^b^	0.8 ± 0.02 ^a^
Gram-negative bacteria		
*Escherichia coli* (ATCC25922)	0.8 ± 0.01 ^b^	1.2 ± 0.03 ^a^
*Klebsiella pneumonia* (ATCC700603)	0.6 ± 0.04 ^b^	1.2 ± 0.01 ^a^

^1^ Diameter includes 5 mm well diameter. MIC—minimum inhibition concentration (mg/mL). Different letters (^a^ and ^b^) indicate significant differences at *p* ≤ 0.05.

**Table 3 marinedrugs-20-00785-t003:** Total phenolic and flavonoid contents of *Codium tomentosum* and *Actinotrichia fragilis*.

Test ^1^	*Codium tomentosum*	*Actinotrichia fragilis*
Total phenolic content (mg/g of extract)	32.28 ± 1.63 ^a^	19.96 ± 1.28 ^b^
Total flavonoid content (mg/g of extract)	4.54 ± 1.48 ^c^	3.86 ± 1.02 ^d^

^1^ Means in the same column followed by different lowercase letters are significantly different (*p* < 0.05).

**Table 4 marinedrugs-20-00785-t004:** Antioxidant activity of *Codium tomentosum* and *Actinotrichia fragilis* as measured by DPPH assay.

Extracts	DPPH (IC_50_) μg/mL
Ascorbic acid	22.71 ± 0.03 ^c^
*Codium tomentosum*	75.32 ± 0.07 ^b^
*Actinotrichia fragilis*	94.43 ± 0.02 ^a^

IC_50_ (μg/mL): Inhibitory concentrations at which 50% of DPPH radicals are scavenged. Means in the same column followed by different lowercase letters are significantly different (*p* < 0.05).

**Table 5 marinedrugs-20-00785-t005:** The influence of various concentrations of *Codium tomentosum* aqueous extract on PBMC viability.

Concentration (µg/mL)	Inhibition %	Viability %
250	69	31
125	69	31
62.5	64	36
31.2	30	70
15.6	29	71
7.8	27	73
IC_50_ = 33.7 ± 1.02 µg/mL		

**Table 6 marinedrugs-20-00785-t006:** The influence of various concentrations of *Actinotrichia fragilis* aqueous extract on PBMC viability.

Concentration (µg/mL)	Inhibition %	Viability %
250	73	27
125	60	40
62.5	47	53
31.2	39	61
15.6	37	63
7.8	11	89
IC_50_ = 51.0 ± 1.14 µg/mL		

**Table 7 marinedrugs-20-00785-t007:** Phenolic compounds of algal extracts identified by HPLC.

Phenolic Compounds	*Codium tomentosum*	*Actinotrichia fragilis*
	Conc. (µg/g)	
Gallic acid	174.32	303.68
Chlorogenic acid	ND	29.50
Catechin	29.92	87.10
Methyl gallate	12.91	0.00
Caffeic acid	70.21	12.79
Syringic acid	50.03	0.00
Pyro catechol	ND	0.00
Rutin	ND	22.73
Ellagic acid	232.69	190.62
Coumaric acid	ND	0.00
Vanillin	ND	3.36
Ferulic acid	26.13	15.99
Naringenin	61.09	23.77
Daidzein	791.39	10.47
Quercetin	48.03	65.78
Cinnamic acid	35.51	6.10
Apigenin	224.88	30.28
Kaempferol	77.28	0.00
Hesperetin	75.90	0.00

ND: Not detected.

**Table 8 marinedrugs-20-00785-t008:** Sensory evaluation of fillet fish fortified with algal extracts.

	Appearance	Consistency	Tenderness	Flavor (Odor and Taste)	Overall Acceptance
T1 (Control)	4.60 ± 1.06 ^a^	5.07 ± 1.03 ^a^	3.53 ± 1.69 ^b^	4.13 ± 1.64 ^a^	4.53 ± 1.18 ^b^
T2 (1% of *C. tomentosum* extract)	5.80 ± 1.15 ^a^	5.67 ± 1.18 ^a^	5.13 ± 1.41 ^a^	5.60 ± 1.63 ^a^	5.93 ± 1.10 ^a^
T3 (1% *A. fragilis* extract)	4.73 ± 1.16 ^a^	4.87 ± 1.19 ^a^	4.80 ± 1.61 ^ab^	4.67 ± 1.11 ^a^	4.80 ± 0.77 ^b^
T4 (0.5% of *C. tomentosum* and 0.5% of *A. fragilis* extracts)	5.13 ± 1.41 ^a^	5.67 ± 1.05 ^a^	5.40 ± 1.59 ^a^	5.33 ± 1.39 ^a^	5.47 ± 1.45 ^a^

Means in the same column followed by different lowercase letters are significantly different (*p* < 0.05).

## Data Availability

Data are available from authors.

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
