# Peer review of "Fillet Fish Fortified with Algal Extracts of Codium tomentosum and Actinotrichia fragilis, as a Potential Antibacterial and Antioxidant Food Supplement"

_marinedrugs, 2022, doi:10.3390/md20120785_

Round 1

Reviewer 1 Report

This is an interesting study on the phenolic characterization and evaluation of antibacterial and antioxidant activities of aqueous extracts from two green and one red seaweeds from the Red Sea. The most novel aspects are related to the fortification of fish fillets to improve nutritional values and sensory evaluation.

The manuscript is clearly written and is easy to follow. The objectives have been reached and the experimental and analytical methodologies are correctly selected and applied. The conclusions are complete and the tables and figures are relevant.

Only minor suggestions:

Probably for the initial general comments of the Introduction other references are vallid and not only those from the authors.

In Table 3, indicate if the content is referref to 1 g extract or to 1 g seaweed. It is in line 163, but tables shoudl be autoexplicative

Check format of Ref. 38, Latin names in italics, coffeic acid; querectin

Author Response

Reviewer (1):

Comment (1):

Probably for the initial general comments of the Introduction other references are valid and not only those from the authors.

Response:

Firstly, we would like to thank the Reviewer for his/her words. They were very valuable in improving the overall quality of the manuscript. We added more references about this specific thematic.

Comment (2):

In Table 3, indicate if the content is referref to 1 g extract or to 1 g seaweed. It is in line 163, but tables shoudl be autoexplicative.

Response:

Firstly, we would like to thank the Reviewer for his/her words. They were very valuable in improving the overall quality of the manuscript. The total phenolic and total flavonoids contents are refered to 1 g extract.

Comment (3):

Check format of Ref. 38, Latin names in italics, coffeic acid; querectin

Response:

Firstly, we would like to thank the Reviewer for his/her words. They were very valuable in improving the overall quality of the manuscript. We revise the references.

Reviewer 2 Report

General comments

The antibacterial and antioxidant capacity of the aqueous extracts of two seaweed species, Codium tomentosum (green algae) and Actinotrichia fragilis (red algae) from the Red Sea, were assessed. Total polyphenol content and total flavonoids were analyzed.  Both aqueous extracts were safe or innocuous, and tested on fish filleted as food supplement and natural preservative. The MS is well written, the terminology used meets accepted standards, and methods are well described. I recommend this manuscript for publication in its almost current state after minor revision. I suggest change the words marine algae for seaweed along the text.  Before the acceptance of this MS all comments and suggestion described below (see specific comments) should be attended.

Specific comments:

Lines 71-72: Please, review this sentence, something is missing.

Lines 83-84: The periods between sentences and references numbers are misplaced.

Line 106: Results showed in this table could be analyzed by t-student test in order to stablish if exist or not statistical differences between means.

Line 112: The same comment as above for the results showed in this table.

Lines 124-152: As demonstrated before in a plethora of published works (for reference see Osuna-Ruiz et al 2016 Pharmaceutical Biology 54(10) 2196-2210), the organic solvent used to perform the extraction exert a particular effect in the chemical composition of the seaweed extract. The chemopreventive activity of seaweed extract is linked to their chemical composition. Thus, all the comparisons made of your own results with published works should be made carefully, always should be indicated the solvent used in order to know if data are really comparable or not. Indeed, I strongly suggest only used published data from aqueous extracts, as you performed.   

Line 169: A reference of published work should be added in order to sustain the sentence.

Lines 182-185: Same as mentioned above.

Lines 193-201: There are several published works in order to better sustain this discussion.

Line 235: Please, add the respectively letter to each chromatogram in order to be identified with the title of this figures.

Lines 227-279: The reproductive stage of seaweeds was recorded? This is particularly important for the red seaweed, and should be mentioned.

Lines 280-286: The seaweed extracts were chemical characterized for the content of polysaccharides? For the temperature and time used during extraction is quite sure that polysaccharides were also extracted, and could explain the chemopreventive activities assessed in your study.  

Line 358-360: It seems you do not have enough data to perform parametric statistics (i.e. ANOVA). Please, verify the properly use of statistical methods.  

Author Response

Reviewer (2):

Comments (1):

General comments

The antibacterial and antioxidant capacity of the aqueous extracts of two seaweed species, Codium tomentosum (green algae) and Actinotrichia fragilis (red algae) from the Red Sea, were assessed. Total polyphenol content and total flavonoids were analyzed.  Both aqueous extracts were safe or innocuous, and tested on fish filleted as food supplement and natural preservative. The MS is well written, the terminology used meets accepted standards, and methods are well described. I recommend this manuscript for publication in its almost current state after minor revision. I suggest change the words marine algae for seaweed along the text.  Before the acceptance of this MS all comments and suggestion described below (see specific comments) should be attended.

Specific comments:

Lines 71-72: Please, review this sentence, something is missing.

Lines 83-84: The periods between sentences and references numbers are misplaced.

Response: Firstly, we would like to thank the Reviewer for his/her words. They were very valuable in improving the overall quality of the manuscript. We revise the question addressed.

Comments (2):

Lines 124-152: As demonstrated before in a plethora of published works (for reference see Osuna-Ruiz et al 2016 Pharmaceutical Biology 54(10) 2196-2210), the organic solvent used to perform the extraction exert a particular effect in the chemical composition of the seaweed extract. The chemopreventive activity of seaweed extract is linked to their chemical composition. Thus, all the comparisons made of your own results with published works should be made carefully, always should be indicated the solvent used in order to know if data are really comparable or not. Indeed, I strongly suggest only used published data from aqueous extracts, as you performed.  

Line 169: A reference of published work should be added in order to sustain the sentence.

Lines 182-185: Same as mentioned above.

Response: We revise the results and discussion section, which we added more information, and supported with new references.

Comments (3):

Lines 193-201: There are several published works in order to better sustain this discussion.

Line 235: Please, add the respectively letter to each chromatogram in order to be identified with the title of this figures.

ResponseWe added more information to the section. Also, we added the letters to the figure.

Comments (4):

Line 106: Results showed in this table could be analyzed by t-student test in order to stablish if exist or not statistical differences between means.

Line 112: The same comment as above for the results showed in this table.

Line 358-360: It seems you do not have enough data to perform parametric statistics (i.e. ANOVA). Please, verify the properly use of statistical methods.  

Response:

Firstly, we would like to thank the Reviewer for his/her words. They were very valuable in improving the overall quality of the manuscript.

We did all the statistical analysis to all the mentioned tables. Statistical significance is illustrated by one-way ANOVA and Fisher’s grouping test and is performed using Minitab® (Version 16) software.

Table 1 Antibacterial potentials of algal extracts.

Strains

Inhibition zone diameter (mm )1

Codium tomentosum

(Green algae)

Actinotrichia fragilis (Red algae)

P-Value

Gram-positive bacteria

Staphylococcus aureus (ATCC25923)

22±0.04a

18±0.05b

0.002

Streptococcus pyogenes (EMCC1772)

20±0.01a

16±0.01b

0.001

Gram-negative bacteria

Escherichia coli (ATCC25922)

14±0.08a

12±0.05b

0.001

Klebsiella pneumonia (ATCC700603)

18±0.02a

14±0.07b

0.003

1Different letters (a and b) indicate significant differences at p ≤ 0.05.

Table 2 Minimum Inhibitory Concentration (MIC) of Codium tomentosum (Green algae) against pathogenic bacteria indicated as a zone of inhibition for each concentration.

Strains

MIC (mg/ml) 1

Codium tomentosum

(Green algae)

Actinotrichia fragilis (Red algae)

P-Value

Gram-positive bacteria

Staphylococcus aureus (ATCC25923)

0.4±0.05b

0.8±0.03a

0.001

Streptococcus pyogenes (EMCC1772)

0.4±0.04b

0.8±0.02a

0.001

Gram-negative bacteria

Escherichia coli (ATCC25922)

0.8±0.01b

1.2±0.03a

0.001

Klebsiella pneumonia (ATCC700603)

0.6±0.04b

1.2±0.01a

0.001

1Different letters (a and b) indicate significant differences at p ≤ 0.05.

Table 8 Sensory Evaluation of Fillet fish fortified with the algal extracts.

Appearance

Consistency

Tenderness

Flavor (odour and taste)

overall acceptance

T1 (Control)

4.60±1.06a

5.07±1.03a

3.53±1.69b

4.13±1.64a

4.53±1.18b

T2 (1% of C. tomentosum extract)

5.80±1.15a

5.67±1.18a

5.13±1.41a

5.60±1.63a

5.93±1.10a

T3 (1% A. fragilis extract)

4.73±1.16a

4.87±1.19a

4.80±1.61ab

4.67±1.11a

4.80±0.77b

T4 (0.5% of C. tomentosum and 0.5% of A. fragilis extracts)

5.13±1.41a

5.67±1.05a

5.40±1.59a

5.33±1.39a

5.47±1.45a

P-value

0.076

0.153

0.021

0.069

0.001

Different letters (a and b) indicate significant differences at p ≤ 0.05.

Comments (5):

Lines 227-279: The reproductive stage of seaweeds was recorded? This is particularly important for the red seaweed, and should be mentioned.  

Response:

Firstly, we would like to thank the Reviewer for his/her words. They were very valuable in improving the overall quality of the manuscript.

The two macroalgal species were collected randomly from the site in the summer seasons. Because of the large amount of collected algae used in the extraction, no reproductive stage was recorded, but this valuable observation will be made in the following work.

Comments (6):

Lines 280-286: The seaweed extracts were chemical characterized for the content of polysaccharides? For the temperature and time used during extraction is quite sure that polysaccharides were also extracted, and could explain the chemopreventive activities assessed in your study.  

Response:

Firstly, we would like to thank the Reviewer for his/her words. They were very valuable in improving the overall quality of the manuscript.

In this study, we do not characterize the polysaccharide content because it requires different extraction methods that need pH adjustments, precipitation with ice… etc. We concentrated here on the polyphenols compounds which are the key antioxidant and antibacterial agents with numerous benefits for disease prevention and human health. Furthermore, our future research will deal with the polysaccharides (free, sulfated polysaccharides, glycolipids, and glycoproteins), which are important compounds with potential biological activities. 

Thank you for your consideration